# Deep Retrieval: An End-to-End Structure Model for Large-Scale Recommendations

## Abstract

One of the core problems in large-scale recommendations is to retrieve top relevant candidates accurately and efficiently, preferably in sub-linear time. Previous approaches are mostly based on a two-step procedure: first learn an inner-product model and then use maximum inner product search (MIPS) algorithms to search top candidates, leading to potential loss of retrieval accuracy. In this paper, we present Deep Retrieval (DR), an end-to-end learnable structure model for large-scale recommendations. DR encodes all candidates into a discrete latent space. Those latent codes for the candidates are model parameters and to be learnt together with other neural network parameters to maximize the same objective function. With the model learnt, a beam search over the latent codes is performed to retrieve the top candidates. Empirically, we showed that DR, with sub-linear computational complexity, can achieve almost the same accuracy as the brute-force baseline.

## 1 Introduction

Recommendation systems have gained great success in various commercial applications for decades. The objective of these systems is to retrieve relevant candidate items from an corpus based on user features and historical behaviors. One of the early successful techniques of recommendation systems is the collaborative filtering (CF), which makes predictions based on the simple idea that similar users may prefer similar items. Item-based collaborative filtering (Item-CF) (Sarwar et al., 2001) extends the idea by considering the similarities between items and items, which lays the foundation for Amazon's recommendation system (Linden et al., 2003).

In the Internet era, the amount of candidates from content platforms and the number of active users in those platforms rapidly grow to tens to hundreds of millions. The scalability, efficiency as well as accuracy are all challenging problems in the design of modern recommendation systems. Recently, vector-based retrieval methods have been widely adopted. The main idea is to embed users and items in a latent vector space, and use the inner product of vectors to represent the preference between users and items. Representative vector embedding methods include matrix factorization (MF) (Mnih & Salakhutdinov, 2008; Koren et al., 2009), factorization machines (FM) (Rendle, 2010), DeepFM (Guo et al., 2017), Field-aware FM (FFM) (Juan et al., 2016), etc. However, when the number of items is large, the cost of brute-force computation of the inner product for all items can be prohibitive. Thus, maximum inner product search (MIPS) or approximate nearest neighbors (ANN) algorithms are usually used to retrieve top relevant items when the corpus is large. Efficient MIPS or ANN algorithms include tree-based algorithms (Muja & Lowe, 2014; Houle & Nett, 2014), locality sensitive hashing (LSH) (Shrivastava & Li, 2014; Spring & Shrivastava, 2017), product quantization (PQ) (Jegou et al., 2010; Ge et al., 2013), hierarchical navigable small world graphs (HNSW) (Malkov & Yashunin, 2018), etc.

Despite their success in real world applications, vector-based algorithms has two main deficiencies: (1) The objective of learning vector representation and learning good MIPS structure are not well aligned for the recommendation task; (2) The dependency on inner products of user and item embeddings might not be sufficient to capture the complicated structure of user-item interactions (He et al., 2017). In order to break these limitations, tree based models (Zhu et al., 2018; 2019; Zhuo et al., 2020), TDM/JDM, have been proposed. These methods use a tree as indices and map each item to a leaf node of the tree. Learning objectives for model parameters and tree structures are well aligned to improve the accuracy. However, the number of parameters in these models are proportional to the

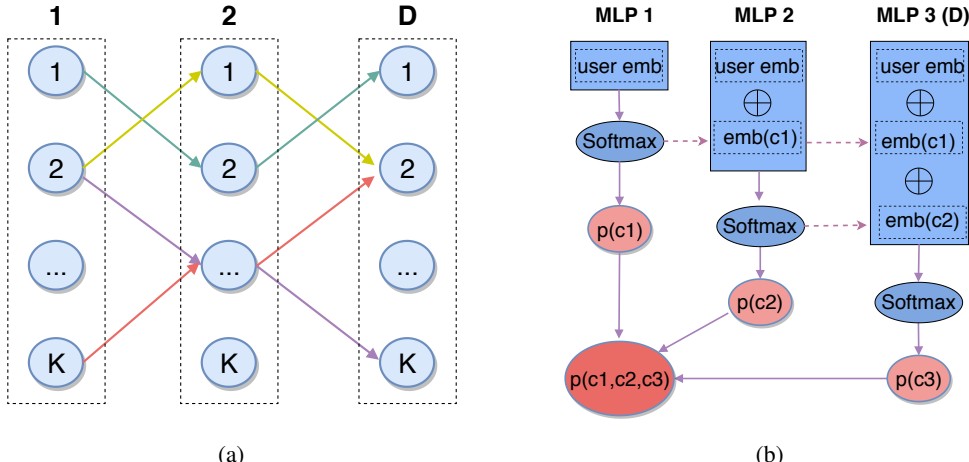

Figure 1: (a) Consider a structure with width $K = 100$ and depth $D = 3$. Assuming an item is encoded by length-$D$ vector $[36, 27, 20]$, which is called a "path". The path denotes that the item is assigned to the $(1, 36), (2, 27), (3, 20)$ indices of the $K \times D$ matrix. In the figure, arrows with the same color form a path. Different paths could intersect with each other by sharing the same index at some layer. (b) Flow chart showing the process for constructing the probability of path $c = [c_1, c_2, c_3]$ given input $x$, $p(c|x, \theta)$.

number of clusters, making the tree structure itself difficult to learn — data available at the leaf level can be scarce and might not provide enough signal to learn a good tree at a finer level.

In this paper, we proposed an end-to-end learnable structure model — Deep Retrieval (DR). In DR, we use a $K \times D$ matrix as in Figure 1a for indexing, motivated by Chen et al. (2018). In this structure, we define a path $c$ as the forward index traverse over matrix columns. Each path is of length $D$ with index value range $\{1, 2, \ldots, K\}$. There are $K^D$ possible paths and each path can be interpreted as a cluster of items. There are two major characteristics in designing the structure. First, There is no "leaf node" as that in a tree, so the data scarcity problem of learning the tree based model can be largely avoided in DR's structure. Second, each item can be indexed by one or more paths — each path could contain multiple items and each item could also belong to multiple paths. This property is naturally accommodated using our probabilistic formulation of the DR model as we will show below. This multiple-to-multiple encoding scheme between items and paths differs significantly with the one-to-one mapping used in tree structure design. In training, the item paths are learnt together with the neural network parameters of the structure model using an expectation-maximization (EM) type algorithm (Dempster et al., 1977). The entire training process is end-to-end and can be easily deployed for large-scale content platforms.

The rest of the paper is organized as follows. In Section 2, we describe the structure model and its structure objective function used in training in detail. We then introduce a beam search algorithm to find candidate paths in the inference stage. In Section 3, we introduce the EM algorithm for training neural network parameters and paths of items jointly. In Section 4, we demonstrate the performance of DR on two public datasets: MovieLens-20M[1] and Amazon books[2]. Experiment results show that DR can almost achieve the brute-force accuracy with sub-linear computational complexity. In Section 5, we conclude the paper and discuss several possible future research directions.

## 2 DEEP RETRIEVAL: AN END-TO-END STRUCTURE MODEL

In this section, we introduce the DR structure model in detail. First, we show how we establish the probability function for user $x$ to select path $c$ given the model parameters $\theta$. This follows the extension to the multi-path mechanism that enables DR to capture multi-aspect properties of items.

---

[1] https://grouplens.org/datasets/movielens
[2] http://jmcauley.ucsd.edu/data/amazon

We then introduce the penalization design that prevents collapsing in allocating items to different paths. Next we describe the beam search algorithm for retrieval. Finally, we present the multi-task joint training procedure of DR with a reranking model.

## 2.1 THE DR MODEL

**The basic model.** The DR model consists of $D$ layers with $K$ nodes, with $DK$ nodes in total. In each layer, we use a multi-layer perceptron (MLP)[3] with skip connections and softmax as output. Each layer takes an input vector and outputs a probability distribution over $\{1, 2, \ldots, K\}$ based on parameters $\theta$. Let $\mathcal{V} = \{1, \ldots, V\}$ be the labels of all items and $\pi : \mathcal{V} \to [K]^D$ be the mapping from items to paths. Assume we are given mapping $\pi$ in this section and will introduce the algorithm for learning the mapping together with $\theta$ in Section 3. In addition, for the mapping $\pi$, we assume an item only belongs to one path and we would extend it to the multi-path setting in next section.

Given a pair of training sample $(x, y)$, which denotes a positive interaction (click, convert, like, etc.) between user $x$ and item $y$, as well as the path $c = (c_1, c_2, \ldots, c_D)$ associated with the item $y$, the probability $p(c|x, \theta)$ is constructed layer by layer as follows (see Figure 1b for a flow chart),

- The first layer takes the user embedding $\mathrm{emb}(x)$ as input, and outputs a probability $p(c_1|x, \theta_1)$ over the $K$ nodes of the first layer, based on parameters $\theta_1$.
- From the second layer onward, we concatenate the user embedding $\mathrm{emb}(x)$ and the embeddings of all the previous layers $\mathrm{emb}(c_{d-1})$ (called path embeddings) as the input of MLP, and output $p(c_d|x, c_1, \ldots, c_{d-1}, \theta_d)$ over the $K$ nodes of layer $d$, based on parameters $\theta_d$.
- The probability of path $c$ given user $x$ is the product of the probabilities of all the layers' outputs.

$$p(c|x, \theta) = \prod_{d=1}^{D} p(c_d|x, c_1, \ldots, c_{d-1}, \theta_d). \tag{1}$$

Given a set of $N$ training samples $\{(x_i, y_i)\}_{i=1}^{N}$, the log likelihood function of the structure model is

$$\mathcal{Q}_{\mathrm{str}}(\theta, \pi) = \sum_{i=1}^{N} \log p(\pi(y_i)|x_i, \theta). \tag{2}$$

The size of the input vector of layer $d$ is the embedding size times $d$, and the size of the output vector is $K$. The parameters of layer $d$ have a size of $\Theta(Kd)$. The parameters $\theta$ contain the parameters $\theta_d$s in all layers, as well as the path embeddings. The parameters in the entire model have an order of $\Theta(KD^2)$, which is significantly smaller than the number of possible paths $K^D$ when $D \geq 2$.

**Multi-path extension.** In tree-based deep models (Zhu et al., 2018, 2019) as well as the structure model we introduced above, each item belongs to only one cluster/path, limiting the capacity of the model to express multi-aspect information in real data. For example, an item related to kebab could belong to a "food" cluster. An item related to flowers could belong to a "gift" cluster. However, an item related to chocolate or cakes could belong to both clusters in order to be recommended to users interested in either food or gifts. In real world recommendation systems, a cluster might not have an explicit meaning such as food or gifts, but this example motivates us to assign each item to multiple clusters. In DR, we allow each item $y_i$ to be assigned to $J$ different paths $\{c_{i,1}, \ldots, c_{i,J}\}$. Let $\pi : \mathcal{V} \to [K]^{D \times J}$ be the mapping from items to multiple paths. Then the multi-path structure objective is straightforwardly defined as,

$$\mathcal{Q}_{\mathrm{str}}(\theta, \pi) = \sum_{i=1}^{N} \log \left( \sum_{j=1}^{J} p(c_{i,j} = \pi_j(y_i)|x_i, \theta) \right). \tag{3}$$

This is because the probability belonging to multiple paths is simply the summation of the probabilities belonging to individual paths.

---

[3] Other neural network architectures such as recurrent neural networks (RNN) can also be applied here. Since $D$ is not very large in our settings, for simplicity, we use MLP.

**Penalization on size of paths.** Directly optimizing $\mathcal{Q}_{\text{str}}(\theta, \pi)$ w.r.t. the item-to-path mapping $\pi$ could fail to ensure to allocate different items into different paths. (We did observe this in practice.) In an extreme case, we could allocate all items into a single path and the probability of seeing this single path given any user $x$ is 1 as in Eq. 1. This is because there is only one available path to chose from. However, this will not help on retrieving the top candidates since there is no differentiation among the items. We must regulate the possible distribution of $\pi$ to be diverse. We introduce the following penalized likelihood function,

$$\mathcal{Q}_{\text{pen}}(\theta, \pi) = \mathcal{Q}_{\text{str}}(\theta, \pi) - \alpha \cdot \sum_{c \in [K]^D} f(|c|) \tag{4}$$

where $\alpha$ is the penalty factor, $|c|$ denotes the number of items allocated in path $c$ and $f$ is an increasing and convex function. A quadratic function $f(|c|) = |c|^2/2$ controls the average size of paths, and higher order polynomials penalize more on larger paths. In our experiments, we use $f(|c|) = |c|^4/4$.

## 2.2 BEAM SEARCH FOR INFERENCE

In the inference stage, we want to retrieve items from the DR model, given user embeddings as input. To this end, we use the beam search algorithm (Reddy et al., 1977) to retrieve top probable paths. In each layer, the algorithm selects top $B$ nodes from the all successors of the selected nodes from the previous layer. Finally it returns $B$ top paths in the final layer. When $B = 1$, this becomes the greedy search. The time complexity of the inference stage is $O(DKB \log B)$, which is sub-linear with respect to the total number of items $V$. The detail of the beam search algorithm is relegated to the appendix A. After the top paths are retrieved, we look up the mapping function $\pi$ to collect the actual items that belong those retrieved paths.

## 2.3 MULTI-TASK LEARNING AND RERANKING WITH SOFTMAX MODELS

The number of items returned by the beam search is much smaller than the total number of items, but often is not small enough to serve the user request, so we would like to rank those retrieved items. However, since each path in DR can contain more than one item, we will not be able to differentiate the items in the same path. We choose to tackle this by jointly training a DR model with a reranker,

$$\mathcal{Q}_{\text{softmax}} = \sum_{i=1}^{N} \log p_{\text{softmax}}(y = y_i | x_i) \tag{5}$$

where $p(y = y_i | x_i)$ is a softmax model with output size $V$. This is trained with the sampled softmax algorithm. The final objective is given by $\mathcal{Q} = \mathcal{Q}_{\text{pen}} + \mathcal{Q}_{\text{softmax}}$. After performing beam search to retrieval a set of candidate items, we rerank those candidates to obtain the final top candidates. Here we use a simple softmax model for this purpose, but we can certainly replace it by a more complex one for better ranking performance.

## 3 LEARNING WITH THE EM ALGORITHM

In the previous section, we introduced the structure model in DR and its objective to be optimized. The objective is continuous with respect to the neural network parameters $\theta$, which can be optimized by any gradient-based optimizer. However, the objective involving the item-to-path mapping $\pi$ is discrete and can not be optimized by a gradient-based optimizer. As this mapping acts as the "latent clustering" of items, this motivates us to use an EM-style algorithm to optimize the mapping and other parameters jointly.

In general, the EM-style algorithm for DR is summarized as follows. At the $t^{\text{th}}$ epoch,

1. E-step: for fixed mapping $\pi^{(t-1)}$, optimize parameter $\theta$ using a gradient-based optimizer to maximize the structure objective $\mathcal{Q}_{\text{pen}}(\theta, \pi^{(t-1)})$.
2. M-step: update mapping $\pi^t$ to maximize the same structure objective $\mathcal{Q}_{\text{pen}}(\theta, \pi)$.

Since the E-step is similar to any standard stochastic optimization, we will focus on the M-step here. For simplicity, we first consider the case where penalization function $f$ is not included. Given a

user-item training pair $(x_i, y_i)$, let the path associated with the item be $(\{\pi_1(y_i), \ldots, \pi_J(y_i)\})$. For a fixed $\theta$, we can rewrite $\mathcal{Q}_{str}(\theta, \pi)$ as

$$\mathcal{Q}_{str}(\theta, \pi) = \sum_{i=1}^{N} \log \left( \sum_{j=1}^{J} p(\pi_j(y_i)|x_i, \theta) \right) = \sum_{v=1}^{V} \left( \sum_{i:y_i=v} \log \left( \sum_{j=1}^{J} p(\pi_j(v)|x_i, \theta) \right) \right),$$

where the outer summation is over all items $v \in \mathcal{V}$ and the inner summation is over all appearances of item $v$ in the training set. We now consider maximizing the objective function over all possible mappings $\pi$. However, for an item $v$, there are $K^D$ number of possible paths so we could not enumerate it over all $\pi_j(v)$'s. We make the follow approximation to further simplify the problem. We use an upper bound $\sum_{i=1}^{N} \log p_i \leq N(\log \sum_{i=1}^{N} p_i - \log N)$ to obtain[4]

$$\mathcal{Q}_{str}(\theta, \pi) \leq \overline{\mathcal{Q}}_{str}(\theta, \pi) = \sum_{v=1}^{V} \left( N_v \log \left( \sum_{j=1}^{J} \sum_{i:y_i=v} p(\pi_j(v)|x_i, \theta) \right) - \log N_v \right),$$

where $N_v = \sum_{i=1}^{N} \mathbb{I}[i : y_i = v]$ denotes the number of occurrences of $v$ in the training set which is independent of the mapping. We define the following score function,

$$s[v, c] \triangleq \sum_{i:y_i=v} p(c|x_i, \theta).$$

Intuitively, $s[v, c]$ can be understood as the importance score of allocating item $v$ to path $c$. In practice, it is impossible to retain all scores as the possible number of paths $c$ is exponentially large, so we only retain a subset of $S$ paths with larger scores through beam search and set the rest of the scores as 0.

Given score $s[v, c]$ estimated, now we consider add back the penalization function. This leads to the following surrogate function in the M-Step,

$$\arg\max_{\{\pi_j(v)\}_{j=1}^{J}} \sum_{v=1}^{V} \left( N_v \log \left( \sum_{j=1}^{J} s[v, \pi_j(v)] \right) - \log N_v \right) - \alpha \cdot \sum_{c \in [K]^D} f(|c|).$$

This can be efficiently solved using a coordinate ascent algorithm and its details can be found in Appendix B.

## 4 EXPERIMENTS

In this section, we study the performance of DR on two public recommendation datasets: MovieLens-20M (Harper & Konstan, 2015) and Amazon books (He & McAuley, 2016; McAuley et al., 2015). We compare the performance of DR with brute-force algorithm, as well as several other recommendation baselines including tree-based models TDM (Zhu et al., 2018) and JTM (Zhu et al., 2019). At the end of this section, we investigate the role of important hyperparameters in DR.

### 4.1 DATASETS AND METRICS

**MovieLens-20M**. This dataset contains rating and free-text tagging activities from a movie recommendation service called MovieLens. We use the 20M subset which were created by the behaviors of 138,493 users between 1995 and 2015. Each user-movie interaction contains a used-id, a movie-id, a rating between 1.0 to 5.0, as well as a timestamp.

In order to make a fair comparison, we exactly follow the same data pre-processing procedure as TDM. We only keep records with rating higher or equal to 4.0, and only keep users with at least ten reviews. After pre-processing, the dataset contains 129,797 users, 20,709 movies and 9,939,873 interactions. Then we randomly sample 1,000 users and corresponding records to construct the validation set, another 1,000 users to construct the test set, and other users to construct the training set.

---

[4]Since this is an upper bound to approximate the true objective we are maximizing, there is no guarantee as to maximizing a surrogate via a lower bound. However we still find it works well in practice.

Table 1: Comparison of precision@10, recall@10 and F-measure@10 for DR, brute-force retrieval and other recommendation algorithms on MovieLens-20M.

| Algorithm | Precision@10 | Recall@10 | F-measure@10 |
|---|---|---|---|
| Item-CF | 8.25% | 5.66% | 5.29% |
| YouTube DNN | 11.87% | 8.71% | 7.96% |
| TDM (best) | 14.06% | 10.55% | 9.49% |
| DR | 20.58% ± 0.47% | 10.89% ± 0.32% | 12.32% ± 0.36% |
| Brute-force | 20.70% ± 0.16% | 10.96% ± 0.32% | 12.38% ± 0.32% |

Table 2: Comparison of precision@200, recall@200 and F-measure@200 for DR, brute-force and other recommendation algorithms on Amazon Books.

| Algorithm | Precision@200 | Recall@200 | F-measure@200 |
|---|---|---|---|
| Item-CF | 0.52% | 8.18% | 0.92% |
| YouTube DNN | 0.53% | 8.26% | 0.93% |
| TDM (best) | 0.56% | 8.57% | 0.98% |
| JTM | 0.79% | 12.45% | 1.38% |
| DR | 0.95% ± 0.01% | 13.74% ± 0.14% | 1.63% ± 0.02% |
| Brute-force | 0.95% ± 0.01% | 13.75% ± 0.10% | 1.63% ± 0.02% |

For each user, the first half of the reviews according to the timestamp are used as historical behavior features and the latter half are used as ground truths to be predicted.

**Amazon books.** This dataset contains user reviews of books from Amazon, where each user-book interaction contains a user-id, an item-id, and the corresponding timestamp. Similar to MovieLens-20M, we follow the same pre-processing procedure as JTM. The dataset contains 294,739 users, 1,477,922 items and 8,654,619 interactions. Please note that Amazon books dataset has much more items but sparser interactions than MovieLens-20M. We randomly sample 5,000 users and corresponding records as the test set, another 5,000 users as the validation set and other users as the training set. The construction procedures of behavior features and ground truths are the same as in MovieLens-20M.

**Metrics.** We use precision, recall and F-measure as metrics to evaluate the performance for each algorithm. The metrics are computed for each user individually, and averaged without weight across users, following the same setting as both TDM and JTM. We compute the metrics by retrieving top 10 and 200 items for each user in MovieLens-20M and Amazon books respectively.

**Model and training.** Since the dataset is split in a way such that the users in the training set, validation set and test set are disjoint, we drop the user-id and only use the behavior sequence as input for DR. The behavior sequence is truncated to length of 69 if it is longer than 69, and filled with a placeholder symbol if it is shorter than 69. A recurrent neural network with GRU is utilized to project the behavior sequence onto a fixed dimension embedding as the input of DR. We adopt the multi-task learning framework, and rerank the items in the recalled paths by a softmax reranker. We train the embeddings of DR and softmax jointly for the initial two epochs, freeze the embeddings of softmax and train the embeddings of DR for two more epochs. The reason is to prevent overfitting of the softmax model. In the inference stage, the number of items retrieved from beam search is not fixed due to the differences of path sizes, but the variance is not large. Empirically we control the beam size such that the number of items from beam search is 5 to 10 times the number of finally retrieved items.

## 4.2 EMPIRICAL RESULTS

We compare the performance of DR with the following algorithms: Item-CF (Sarwar et al., 2001), YouTube product DNN (Covington et al., 2016), TDM and JTM. We directly use the numbers of Item-CF, Youtube DNN, TDM and JTM from TDM and JTM papers for fair comparison. Among the different variants of TDM presented, we pick the one with best performance. The result of JTM is

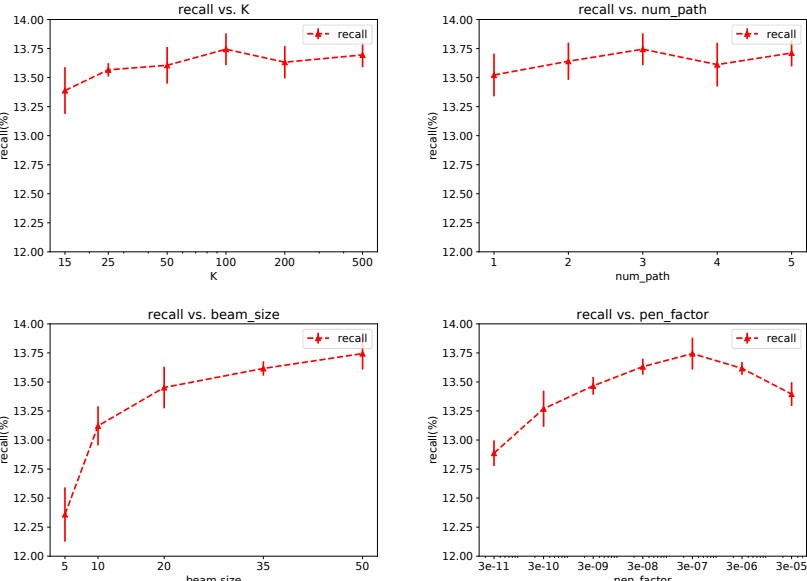

Figure 2: Relationship between recall@200 in Amazon Books experiment and model width $K$, number of paths $J$, beam size $B$ and penalty factor $\alpha$, respectively.

only available for Amazon books. We also compare DR with brute-force retrieval algorithm, which directly computes the inner-product of user embedding and all the item embeddings learnt in the softmax model and returns the top $K$ items. The brute-force algorithm is usually computationally prohibitive in practical large recommendation systems, but can be used as an upper bound for small dataset for inner-product based models.

Table 1 shows the performance of DR compared to other algorithms and brute-force for MovieLens-20M. Table 2 shows the results for Amazon books. For DR and brute-force, we independently train the same model for 5 times and compute the mean and standard deviation of each metric. We conclude the following results.

- DR performs better than other methods including tree-based retrieval algorithms such as TDM and JTM.
- The performance of DR is very close to or on par with the performance of brute-force method, which is an upper bound of the performance of vector-based methods such as Deep FM and HNSW. However, the inference speed of DR is 4 times faster than brute-force in Amazon books dataset (see Table 3 in Appendix C.1).

### 4.3 SENSITIVITY OF HYPERPARAMETERS

DR introduces some key hyperparameters which may infect the performance dramatically, including the width of the structure model $K$, depth of model $D$, number of multiple paths $J$, beam size $B$ and penalty factor $\alpha$. In the MovieLens-20M experiment, we choose $K = 50$, $D = 3$, $B = 25$, $J = 3$ and $\alpha = 3 \times 10^{-5}$. In the Amazon books experiment, we choose $K = 100$, $D = 3$, $B = 50$, $J = 3$ and $\alpha = 3 \times 10^{-7}$. Using the Amazon books dataset, we show the role of these hyperparameters and see how they may affect the performance. We present how the recall@200 change as these hyperparameters change in Figure 2. We keep the value of other hyperparameters unchanged when varying one hyperparameter. Precision@200 and F-measure@200 follow similar trends so we plot them in the appendix.

- **Width of model** $K$ controls the overall capacity of the model. If $K$ is too small, the number of clusters is too small for all the items; if $K$ is too big, the time complexity of training and inference stages grow linearly with $K$. Moreover, large $K$ may increase the possibility of overfitting. An appropriate $K$ should be chosen depending on the size of the corpus.

- **Depth of model** $D$. Using $D = 1$ is obviously not a good idea since it fails to capture dependency among layers. In Tabel 4 in Appendix C.2. we present the result for $D = 2, 3$ and 4 for Amazon books dataset, and conclude that $D = 3$ is a good choice as a trade-off between model performance and memory usage in both experiments.

- **Number of paths** $J$ enables the model to express multi-aspect information of candidate items. The performance is the worst when $J = 1$, and keeps increasing as $J$ increases. Large $J$ may not affect the performance, but the time complexity for training grows linearly with $J$. In practice, choosing $J$ between 3 and 5 is recommended.

- **Beam size** $B$ controls the number of candidate paths to be recalled. Larger $B$ leads a better performance as well as heavier computation in the inference stage. Notice that greedy search is a special case when $B = 1$, whose performance is worse than beam search with larger $B$.

- **Penalty factor** $\alpha$ controls the number of items in each path. The best performance is achieved when $\alpha$ falls in a certain range. Smaller $\alpha$ leads to a larger path size (see Table 5 in Appendix C.3) hence heavier computation in the reranking stage. Beam size $B$ and penalty factor $\alpha$ should be appropriately chosen as a trade off between model performance and inference speed.

Overall, we can see that DR is fairly stable to hyperparameters since there is a wide range of hyperparameters which leads to near-optimal performances.

## 5 CONCLUSION AND DISCUSSION

In this paper, we have proposed Deep Retrieval, an end-to-end learnable structure model for large-scale recommender systems. DR uses an EM-style algorithm to learn the model parameters and paths of items jointly. Experiments have shown that DR performs well compared with brute-force baselines in two public recommendation datasets.

There are several future research directions based on the current model design. Firstly, the structure model defines the probability distribution on paths only based on user side information. A useful idea would be how to incorporate item side information more directly into the DR model. Secondly, in the structure model, we only make use of positive interactions such as click, convert or like between user and item. Negative interactions such as non-click, dislike or unfollow should also be considered in future work to improve the model performance. Finally, we currently use the simple softmax model as a reranker and we plan to use a more complex model in the future.

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
