# OpenReview forum: "Deep Retrieval: An End-to-End Structure Model for Large-Scale Recommendations"
_ICLR.cc/2021/Conference — Reject_

### Official Review · AnonReviewer4 · 2020-10-27
**Novel attempt without clear motivation and convincing experiments**

**Rating:** 4
**Confidence:** 4

**Review:**

The paper seeks to propose an end-to-end learnable retrieval model for recommendation, to replace existing two-step based approaches (learn embeddings first, then do MIPS search).

The underlying model structure is motivated by KD-code (Chen et al. 2018), where each item is encoded as a D-dimensional discrete vector (with a cardinality K in each dim). Then a conditional, probabilistic framework is proposed to learn the model with a multi-path extension for further improvement. After training, Beam Search is adopted to retrieve top candidates.

The paper is technically sound, and it's new to adopt a KD-code like model for retrieval. However, I have some concerns in motivation, methods, and experiments.

- The significant feature of the DR model (from the claim in the abstract) to encode all candidates in a discrete latent space. However, there are some previous attempts in this direction that are not discussed. For example VQ-VAE[1] also learns a discrete space. Another more related example is HashRec[2], which (end-to-end) learns binary codes for users and items for efficient hash table retrieval. It's not clear of the connections and why the proposed discrete structure is more suitable.

- The experiments didn't show the superiority of the proposed method. As a retrieval method, the most common comparison method (e.g. https://github.com/erikbern/ann-benchmarks) is the plot of performance-retrieval time, which is absent in this paper. The paper didn't compare the efficiency against the baselines like TDM, JTM, or ANN-based models, which makes the experiments less convincing as the better performance may due to the longer retrieval time.

- It's not clear to me what retrieval/MIPS search methods are adopted for Item-CF, Youtube DNN.

- What's the performance of purely using softmax?

- It seems only DR uses RNNs for sequential behavior modeling, while the baselines didn't. This'd be a unfair comparison, and sequential methods should be included if DR uses RNN and sequential actions for training.

- I didn't understand the motivation of using the multi-path extension. As you already encode each item in D different clusters, this should be enough to express different aspects with a larger D. Why a multi-path variant is needed for making the model more expressive?

- The Beam Search may not guarantee sub-linear time complexity due to the new hyper-parameter B. It's possible that a very large B is needed for retrieving enough candidates.

In summary, it's not clear to me why the proposed discrete structure is more suitable for the task given we have Tree-based and binary code based approaches (that are also end-to-end learnable). And the experiments didn't show the superiority of the proposed method due to the lack of important comparisons (retrieval time, against HashRec, etc.).

[1]Neural Discrete Representation Learning, NIPS'17
[2]Candidate Generation with Binary Codes for Large-scale Top-N Recommendation, CIKM'19

---

### Official Review · AnonReviewer3 · 2020-10-28
**Interesting algorithm, but sub-par experimentation protocol**

**Rating:** 3
**Confidence:** 5

**Review:**

##########################################################################
Summary:

This paper presents an end-to-end deep retrieval method for recommendation. The model encodes all candidates into a discrete latent space, and learns the latent space parameters alongside the other neural network parameters. Recommendation is performed through beam search. The paper compares the method on two public dataset against several methods (DNN, CF, TDM) and concludes that it can achieve the same result as a brute-force solution in sub-linear time.

##########################################################################
Reasons for score:

The paper presents an interesting end-to-end deep retrieval approach.  However, the paper suffers from several key limitations:

First, it makes a very strong assumption (that vector-based approaches have fundamental limitations).  Because this assertion is very strong, it should be backed by a more thorough analysis than what is done on the paper (more details below).

Second, it fails to take into account several key state-of-the-art methods (such as VAE).  The method proposed in the paper might perform significantly worse than this SOTA based on their reported results.

Finally, it brings confusion between two problems: the one of choosing an algorithm (vector-based versus deep end-to-end) and the one of choosing a brute-force vs approximate nearest-neighbor.  Yet it is well-known that approximate nearest neighbor search is often almost as good as brute-force nearest neighbor, as can be seen here: http://ann-benchmarks.com/

The method presented in the paper is interesting, though.  I believe this work could be published, but with significantly more research work.

##########################################################################
Pros:

- Novelty: this paper present a method that, as far as I know, is novel.

- Comparison to tree-based methods: the paper presents an interesting comparison with tree-based approaches

##########################################################################
Cons:

- Lack of validation: first and foremost, the work presented in this paper lacks validation experiments.  For a paper presenting a new algorithm and making a strong claim regarding vector-based methods, we would expect at least 4-5 datasets as is commonly done in the literature, e.g. with MSD, Netflix, Medium, Amazon, Yahoo datasets.  We would also expect more metrics, and in particular, the right recall values as is commonly done in the field.  In particular, other papers use recall@20 and recall@50 (as can be seen in paperswithcode.com) instead of recall@10.

- Missing state-of-the-art: the paper misses on significant portions of state of the art regarding the evaluation.  Several key methods should be included in the evaluation, such as VAEs [1], EASE [2], RACT [3], SLIM [4] and CML [5].  While these methods are not end-to-end, the paper should compare its performance against these methods to conclude whether end-to-end deep retrieval yields better (or even similar) performance compared to them.  It turns out that some of these methods perform well on recall@20 and maybe better than the method presented in this paper.  Note that some of these methods are also sub-linear in the number of items, such as CML.

- The paper is also missing a reference on solving the vector-based limitations, with Off-Policy Learning in Two-Stage Recommender Systems [6].

- The paper claims to address "large-scale" recommender systems (at several places in the paper) but does not address this aspect.  There exist a significant body of literature on the topic of recommender systems operating at the scale of billions of users and items now, e.g. [7], [8].  Working at the scale of MovieLens and AmazonBooks is not large-scale.  In addition, a complexity analysis of the method would be very welcome.

- Lack of clarity: The clarity of the paper could be greatly improved by putting the description of the algorithm in a single place.  At the moment, it is spread between Section 1 and Section 2.1.  In particular, Section 1 introduces D and K but does not explain what they are. Some aspects of the algorithms are described in a single line (a GRU is used to project the behavior sequence, but nothing is explained about it).

- Lack of code: the paper does not provide the code, which does not help for reproducibility and sharing with the community.  Providing code is paramount when proposing a new algorithm.

[1] Daeryong Kim and Bongwon Suh. 2019. Enhancing VAEs for Collaborative Filtering: Flexible Priors Gating Mechanisms. In Proceedings ofthe 13th ACM Conference on Recommender Systems (RecSys ’19). Association for Computing Machinery, New York, NY, USA, 403–407. https://doi.org/10.1145/3298689.3347015

[2] Harald Steck. 2019. Embarrassingly Shallow Autoencoders for Sparse Data. In The World Wide Web Conference (WWW ’19). Association forComputing Machinery, New York, NY, USA, 3251–3257. https://doi.org/10.1145/3308558.3313710

[3] Sam Lobel, Chunyuan Li, Jianfeng Gao, and Lawrence Carin. 2020. RaCT: Toward Amortized Ranking-Critical Training For Collaborative Filtering. InEighth International Conference on Learning Representations (ICLR). https://www.microsoft.com/en-us/research/publication/ract-toward-amortizedranking-critical-training-for-collaborative-filtering/

[4] Xia Ning and George Karypis. 2011. SLIM: Sparse Linear Methods for Top-N Recommender Systems. In Proceedings of the 2011 IEEE 11th InternationalConference on Data Mining (ICDM ’11). IEEE Computer Society, USA, 497–506. https://doi.org/10.1109/ICDM.2011.134

[5] Cheng-Kang Hsieh, Longqi Yang, Yin Cui, Tsung-Yi Lin, Serge Belongie, and Deborah Estrin. 2017. Collaborative Metric Learning. In Proceedings ofthe 26th International Conference on World Wide Web (WWW ’17). International World Wide Web Conferences Steering Committee, Republic andCanton of Geneva, CHE, 193–201. https://doi.org/10.1145/3038912.3052639

[6] Jiaqi Ma, Zhe Zhao, Xinyang Yi, Ji Yang, Minmin Chen, Jiaxi Tang, Lichan Hong, and Ed H. Chi. 2020. Off-Policy Learning in Two-Stage Recommender Systems. In Proceedings of The Web Conference 2020 (WWW ’20). Association for Computing Machinery, New York, NY, USA, 463–473. https://doi.org/10.1145/3366423. 3380130

[7] Chantat Eksombatchai, Pranav Jindal, Jerry Zitao Liu, Yuchen Liu, Rahul Sharma, Charles Sugnet, Mark Ulrich, and Jure Leskovec. 2018. Pixie: A System for Recommending 3+ Billion Items to 200+Million Users in Real-Time. In Proceedings of the 2018 World Wide Web Conference (WWW ’18). International World Wide Web Conferences Steering Committee, Republic and Canton of Geneva, CHE, 1775–1784. https://doi.org/10.1145/3178876.3186183

[8] JizheWang, Pipei Huang, Huan Zhao, Zhibo Zhang, Binqiang Zhao, and Dik Lun Lee. 2018. Billion-scale Commodity Embedding for E-commerce Recommendation in Alibaba. In Proceedings of the 24th ACM SIGKDD International Conference on Knowledge Discovery & Data Mining (KDD ’18). ACM, New York, NY, USA, 839–848. https://doi.org/10.1145/3219819.3219869

#########################################################################
Some typos:

"grow" on page 1

"from the all successors" on page 4

upper-case are missing in the references (e.g. ALSH)

---

### Official Review · AnonReviewer2 · 2020-10-28

**Rating:** 5
**Confidence:** 3

**Review:**

In this paper, the authors proposed Deep Retrieval, which encodes all items in a discrete latent space for end-to-end retrieval. The proposed method is claimed to perform on par with the brute-force method, under sub-linear computational complexity.

1. The authors mainly claim that the objective of learning vector representation and good inner-product search are not well aligned, and the dependence on inner-products of user/item embeddings may not be sufficient to capture their interactions. This is an ongoing research discussion on this domain. I'd recommend the authors to refer to a recent paper, proposing the opposite direction from this submission:
Neural Collaborative Filtering vs. Matrix Factorization Revisited (RecSys 2020) https://arxiv.org/abs/2005.09683

2. According to Figure 1, a user embedding is given as an input, and the proposed model outputs probability distribution over all possible item codes, which in turn interpreted as items. That being said, it seems the user embedding is highly important in this model. A user can be modeled in a various ways, e.g., as a sequence of items consumed, or using some meta-data. If the user embeddings are not representative enough, the proposed model may not work, and on the other hand, if the user embedding is strong, it will estimate the probs more precisely. We would like to see more discussion on this.

3. In the experiment, there are multiple points that can be addressed. (a) Related to the point #2, the quality of embeddings is not controlled. Thus, comparing DR against brute-force proves that the proposed method is effective on MIPS, but not on the end-to-end retrieval. Ideally, we'd like to see experiments with multiple SOTA embeddings to see if applying DR to those embeddings still improves end-to-end retrieval performance. See examples below. (b) The baselines used in the experiment are not representing the current SOTA. Item-based CF is quite an old method, and YouTube DNN is not fully reproducible due to the discrepancy on input features (which are not publicly available outside of YouTube). We recommend comparing against / using embeddings of LLORMA (JMLR'16), EASE^R (WWW'19), and RecVAE (WSDM'20). (c) Evaluation metrics are somewhat arbitrary. The authors used only one k for P@k, R@k, and F1@k, arbitrarily chosen for each dataset. This may look like a cherry-picking, so we recommend to report scores with multiple k, e.g., {1, 5, 10, 50, 100}. Taking a metric like MAP or NDCG is another option.

4. The main contribution of this paper seems faster retrieval on MIPS. Overall, the paper is well-written. We recommend adding more intuitive description why the proposed mathematical form guarantees / leads to the optimal / better alignment to the retrieval structure. That is, how/why the use of greedy search leads to the optimal selection of item codes.

---

### Official Review · AnonReviewer1 · 2020-10-30
**Algorithmic approach for top-k retrieval**

**Rating:** 4
**Confidence:** 3

**Review:**

The paper presents a method for "end-to-end" learning for retrieving top-k items in recommendation system setup. This is achieved by learning the hidden representations of the items and under neural network as a single objective function optimized using the expectation maximization framework. It is claimed that the proposed method achives better results than state-of-the-art tree-based models and those approaches which learn these components separately.

Some of the concerns regarding the paper are as follows :
- The paper lacks a motivation for using the proposed scheme. It says that for tree-based models, the number of parameters is proportional to the number of clusters and hence it is a problem. This is not clear why this is such a problem. Successful application of tree-structure for large-scale problem has been demonstrated in [1,2]. Also, it is not clear how the proposed method addresses data scarcity, which according to the paper happens only in tree-based methods, and not in the proposed method as there are no leaves.

- It is not clear how the proposed structure model (of using K \times D matrix) is different from the Chen etal 2018. The differences and similarities compared to this work should be clearly specified. Also, what seems to be missing is why such an architecture of using stacked multi-layer perceptrons should lead to better performance especially in positive data-scarcity situations where most of the users 'like' or 'buy' only few items.

- The experimental comparison looks unclear and incomplete. The comparison should also be done with the approach proposed in Zhou etal 2020 ICML paper. At the end of Page 6 it is said that the results of JTM were only available for Amazon Books. How do you make sure that same training and test split (as in JTM) is used as the description says that test and validation set is done randomly. Also, it would also be good to see the code and be able to reproduce the results.

- References of some key papers are based on arxiv versions, such as Chen etal 2018 and Zhou etal. 2020, where both the papers have been accepted in ICML conference of respective years.

[1] Extreme Classification in Log Memory using Count-Min Sketch: A Case Study of Amazon Search with 50M Products, NeurIPS 2019

[2] AttentionXML: Label Tree-based Attention-Aware Deep Model for High-Performance Extreme Multi-Label Text Classification, NeurIPS 2019

---

### Decision · Program_Chairs · 2021-01-07
**Final Decision**

**Decision:**

Reject

**Comment:**

The introduced method is novel and interesting. However, as pointed in the reviews the` paper misses several important references. The authors should extend their discussion on related work by methods from both recommender systems and extreme classification. Besides the papers listed by the reviewers, the introduced method seems also to be related to LTLS (https://arxiv.org/abs/1611.01964) and W-LTST (http://papers.neurips.cc/paper/7953-efficient-loss-based-decoding-on-graphs-for-extreme-classification.pdf), as well as to probabilistic classifier chains (https://icml.cc/Conferences/2010/papers/589.pdf) used for multi-label classification (recommendation can be reduced to multi-label classification under 0/1 loss by coding each item using a binary code of a fixed length). Nevertheless, the introduced method seems to be novel, nicely reusing and fitting together existing ideas.

Unfortunately, the authors did not submit any rebuttal. Therefore, the paper cannot be accepted to ICLR. We encourage the authors to work further and extend the paper by an exhaustive discussion about related work, a wider experimental study, a more detailed description of all the steps of the method.